# Current Pattern of Psychiatric Comorbidity and Psychotropic Drug Prescription in Child and Adolescent Patients

**DOI:** 10.3390/medicina55050159

**Published:** 2019-05-17

**Authors:** Mengühan Araz Altay, Leyla Bozatlı, Begüm Demirci Şipka, Işık Görker

**Affiliations:** Department of Child and Adolescent Psychiatry, Trakya University School of Medicine, 22030 Edirne, Turkey; leylyabozatli@gmail.com (L.B.); begumdemirci55@gmail.com (B.D.Ş.); isikgorker@gmail.com (I.G.)

**Keywords:** child and adolescent psychiatry, outpatient treatment, symptoms, DSM-5, psychotropic medication

## Abstract

*Background:* In recent years, patterns of the use of psychotropic drugs vary with increasing rates of psychiatric presentation and diagnosis in children and adolescents. *Purpose:* In this study, we aimed to investigate distributions of current psychiatric symptoms and diagnosis, patterns of the use of psychotropic drugs, and differences according to age and gender in patients presented to a child and adolescent outpatient clinic. *Methods:* All patients aged between 0 and 18 years presenting to a child and adolescent psychiatry outpatient clinic between November 1, 2017 and November 1, 2018 were included in the study. Files of all patients were examined in detail, and patients’ demographic characteristics, symptoms, psychiatric diagnoses established according to the fifth edition of the Diagnostic and Statistical Manual of Mental Disorders (DSM-5), psychotropic drugs initiated, and side effect profiles were recorded. Psychiatric symptoms and diagnostic features of the patients were determined, and the differences were investigated according to gender. Clinical characteristics were compared between diagnosed and undiagnosed patients, and between patients with and without drug initiation. *Results:* Of the 2066 patients, 1298 (62.8%) were male and the mean age was 10.14 ± 4.42 years. The most common symptoms were hyperactivity (23.8%) and inattention (21.6%) in males, inattention (15.1%) and irritability (14.2%) in females, and 79% of the patients received one or more psychiatric diagnoses. The most common psychiatric diagnoses in both genders were attention-deficit hyperactivity disorder (ADHD), specific learning disorder (SLD), and conduct disorder, respectively. Of the patients who received a psychiatric diagnosis, 61.8% were using psychotropic drugs, with the majority of them (71.3%) receiving monotherapy. The most frequently initiated drugs included psychostimulants, antipsychotics, and antidepressants, with 28.7% of the drug user patients receiving multiple drug therapy. *Conclusion:* Our study indicates that rate of presentation to child and adolescent psychiatry outpatient clinics is increasing, and rates of diagnosis and initiation of psychiatry drugs are high among the presented children. The prevalence of ADHD shows an increase in males and females in our country, and psychiatric polypharmacy has reached significant rates.

## 1. Introduction

Childhood and adolescence is a period where both physical and psychological rapid changes occur. It is reported that the incidence of psychiatric diseases is increasing in children and adolescents in recent years. The overall prevalence was found to be 13.4% in a recent meta-analysis with adolescent data from 27 countries [1]. The incidence of mental disorders in children and adolescents is reported to be between 7% and 35% in developing countries [2,3]. Psychiatric diseases seen in this period affect both the children and the family. In addition, it is known that many diseases emerging in adults have grounds in childhood [4]. Therefore, diagnosis and treatment of psychiatric disorders in this period affect functionality both in childhood and adult periods.

The common psychiatric disorders such as mood, stress related, and somatoform disorders affecting adults also affect many children and adolescents [3,4]. However, attention-deficit hyperactivity disorder (ADHD), specific learning disorder (SLD), autism spectrum disorder (ASD), conduct disorder, and elimination disorders such as enuresis and encopresis are seen especially in children and adolescents [5,6]. Recently, diagnostic criteria of many psychiatric disorders were modified in the fifth edition of the Diagnostic and Statistical Manual of Mental Disorders (DSM-5) [6]. Changes in diagnostic criteria and physician and public awareness works have led to the increase in presentations to child and adolescent psychiatry clinics in recent years. Decreased communication with social media coming into public life has also played a role in the increase of these presentations. As a result of these, increases have been reported in the diagnosis rate of many childhood mental disorders, particularly ADHD and ASD [7].

Turkey has a growing young population. Turkey’s health ministry for the last 10 years has primarily served awareness activities for psychiatric disorders such as ASD and SLD. There were also significant increases in the number of child and adolescent psychiatrists. As a result of these, the number of appointments to child and adolescent psychiatry has been increased in recent years. In Turkey, the use of psychotropic drugs in children and adolescents was reported to be 23.4% in a study conducted in 1992, 24.1% in 2005, and 43.4% in 2014 [8,9,10].

The frequency and type of psychiatric disorder occurring during childhood and adolescence varies according to the gender. In children and adolescents, studies have shown that conduct disorders and ADHD are more common in males, and anxiety and depressive disorders are more common in females [11]. The presence of psychiatric comorbidities complicates the diagnostic process, which has a significant impact on the treatment of primary psychiatric disorder [12]. Also, approximately 16% of adolescents reported two or more comorbid psychiatric disorders [13].

Current studies show increases in physicians’ choice of psychotropic medical therapy in children and adolescents and polypharmacy rates in childhood [14,15]. In parallel with the increase in the rate of psychiatric diagnoses and appointments to child and adolescent psychiatry, there are changes in drug use patterns in adolescents. New studies are needed to demonstrate these changes. In the light of these studies, clinicians can review themselves.

In this study, we aimed to evaluate children and adolescents presenting to a child and adolescent psychiatry outpatient clinic of a high-volume tertiary healthcare center within one year, and to investigate the distribution of patients’ symptoms and diagnoses according to DSM-5 criteria, current psychotropic drug profiles, and differences according to age and gender.

## 2. Materials and Methods

This study was conducted in the Child and Adolescent Psychiatry Clinic of Trakya University School of Medicine, which is a clinic with the highest patient volume in the Trakya region of Turkey. The study was designed as a retrospective cohort study, and all patients aged between 0 and 18 years presenting to the Child and Adolescent Psychiatry Outpatient Clinic between November 1, 2017 and November 1, 2018 were included in the study. A total of 2066 patients were included in the study between these dates. Files of all patients were examined in detail, and patients’ demographic characteristics, symptoms, psychiatric diagnoses, psychotropic drugs initiated, and side effect profiles were recorded. Psychiatric diagnoses of the patients were based on DSM-5 criteria using clinician’s opinion and supportive assessment tools. For example, these assessment tools included the Modified Checklist for Autism in Toddlers (M-CHAT) in ASD scanning in toddlers, the Autism Behavior Checklist (ABC) in different forms and level of severity of ASD, Turgay’s ADHD Evaluation Scale for ADHD, the Specific Learning Difficulties Symptom Scale for SLD, the State-Trait Anxiety Inventory (STAI) for anxiety disorders, and the Beck Depression Inventory (BDI) for depressive disorders [6,16,17,18,19,20,21]. The study protocol was approved by Trakya University Scientific Research Ethics Committee dated 05/11/2018 and numbered TUTF-BAEK 2018/381.

### Statistical Analysis

Data were analyzed by Statistical Package for Social Sciences (SPSS) version 17.0 for Windows (IBM, Armonk, NY, USA). Whether or not the distributions of continuous variables were normal was determined by the Kolmogorov–Smirnov test. Data were shown as mean ± standard deviation for continuous variables. Number of cases and percentages were used for categorical data. Mean differences between continuous measures were compared by Student’s *t*-test, whereas the Mann–Whitney U test was applied for comparisons of the not normally distributed data. Categorical variables were analyzed by chi-square or Fisher’s exact test, where applicable. Statistical significance was accepted as *p* < 0.05.

## 3. Results

Of the 2066 patients, 1298 (62.8%) were male and the mean age was 10.14 ± 4.42 years. Of all patients, 22.6% were in 0–6 years, 45.5% in 7–11 years, and 31.9% in 12–18 years age groups. The mean number of psychiatric outpatient visits of the patients within one year was found as 4.0 ± 3.0. Of the patients, 45.6% had one symptom, while 22.7% had two, and 14.5% had three symptoms. As the result of psychiatric assessment, 21% of the patients were not diagnosed with any psychiatric disorder, 51.4% received only one diagnosis, and 27.7% received more than one psychiatric diagnosis. One or more psychotropic therapies were initiated in 49.1% of the patients.

The most common symptoms of the patients for presentation to the child and adolescent clinic were inattention (19.2%), hyperactivity (17.9%), irritability (13.6%), and naughtiness (11.6%). Symptoms were analyzed according to gender; inattention, hyperactivity, naughtiness, speech delay, language impairment (stuttering, articulation disorder), school-teacher problems, tic disorder, inability to make eye contact, and not responding to being called were significantly higher in males, while various fears (exam, claustrophobia, loneliness etc.), self-mutilation and suicide attempts, physical complaints (nausea, vomiting, fainting, abdominal pain etc.), unhappiness and anhedonia, sleep disturbance, habits (finger sucking, hair pulling, nail biting), eating disorders, and boredom were significantly higher in females. No significant difference was found in other symptoms, as shown in Table 1.

Psychiatric diagnoses of the patients were examined; 79% of the patients received psychiatric diagnosis with the most common diagnoses being ADHD (39.1%), SLD (10.2%), and conduct disorder (7.2%), while 21% of the patients were not diagnosed with any psychiatric disorder. Psychiatric diagnoses were analyzed according to gender; ADHD, SLD, ASD, oppositional defiant disorder (ODD), and motor disorder were significantly higher in males, while anxiety disorder (e.g., specific phobia, generalized anxiety disorder, social anxiety disorder), trauma and stressor-related disorder (e.g., posttraumatic stress disorder, acute stress disorder, adjustment disorder, reactive attachment disorder), depressive disorders, somatic symptoms and related disorders, impulse control disorder, adolescent problems, and eating disorders were higher in females, as shown in Table 2.

Of the patients with ADHD, which was the most commonly diagnosed psychiatric disorder; 61.5% were in the combined, 32.5% in attention deficit, and 5.9% in impulsivity subgroups. The most commonly associated psychiatric diagnoses were found as SLD and ADHD. Ninety percent of SLD cases were accompanied by ADHD, while 23.5% of ADHD cases were accompanied by SLD. In our study, diagnosis of accompanying epilepsy was found in 15 patients, while the most commonly accompanying psychiatric diagnosis was ADHD by 73.3% in these patients.

Symptoms were analyzed according to the age groups; the most common symptoms were speech delay and naughtiness in the 0–6 age group, hyperactivity and inattention in the 7–11 age group, and inattention and irritability in the 12–18 age group. Additionally, 38.6% of presentations in the 0–6 age group did not receive a psychiatric diagnosis, with the most common diagnosis being global developmental delay. The most commonly established diagnosis was ADHD in the 7–11, and 12–18 age groups. There were similar rates of patients who were not diagnosed with a psychiatric disorder, as shown in Table 3.

Patients (433) who did not receive a psychiatric diagnosis were examined; the mean age was 8.5 ± 4.6 years, 55.4% were males. The most common symptoms in this group were determined as naughtiness (11.8%), inattention (10.9%), and timidness (6.7%).

Patients with and without psychiatric diagnosis were analyzed; mean age of the diagnosed patients was significantly higher (10.5 ± 4.3 vs. 8.5 ± 4.6, *p* < 0.001). Inattention, hyperactivity, poorer school performance, reading, writing, maths problems, speech delay, disobedience, language impairment, self-mutilation, and disruptive behavior were significantly higher in the diagnosed patients, while sibling jealousy, sleep disturbance, and physical complaints were significantly higher in the undiagnosed patients, as shown in Table 4.

Children (1633) diagnosed at outpatient visits were analyzed; drug therapy was initiated in 61.8% (*n* = 1009) and was not initiated in 38.2% (*n* = 624) of these children. Patients who received and not receive drug therapy were analyzed; age of the drug therapy receivers was older, and number of the presentations within 1 year was higher. ADHD, conduct disorder, SLD, and ASD were significantly higher in patients who received drug therapy, while anxiety disorder, trauma related disorders, and communication disorders were higher in patients who did not receive drug therapy. Drug therapy receivers were compared as patients receiving monotherapy and polytherapy; age groups were similar, and male gender predominance and number of presentations were higher in patients who received polytherapy. ADHD, conduct disorder, ODD, and intellectual disability were higher, while anxiety and trauma-related disorders were lower in patients who received polytherapy, as shown in Table 5.

Among monotherapies; the most commonly used drugs were long-acting psychostimulants by 29.3%, antipsychotics by 24% (risperidone 73.5%, aripiprazole 17.4%, quetiapine 3.7%), antidepressants by 16% (fluoxetine 49.1%, sertraline 36.6%, escitalopram 8.1%), and atomoxetine by 9%.

Of the patients who received drugs, 28.7% were using a combination of more than one drug. Among these, the most commonly used combination was psychostimulant + antipsychotic (12%) and antipsychotic + atomoxetine (4.4%), while the other combinations were rarely preferred, as shown in Table 6.

According to age groups, psychotropic drug therapy was administered in 14.6% of the 0–6 age group, 56.5% of the 7–11 age group, and 62.8% of the 12–18 age group. Drug side effects were observed in 15.5% (*n* = 157) of patients who received drug therapy. The most common side effects were loss of appetite (4.8%), sedation (1.9%), sleep withdrawal (1.3%), irritability (1.2%), and nausea (1.1%).

## 4. Discussion

Our study revealed current symptoms, psychiatric diagnosis, and patterns of use of psychotropic drugs in patients presenting to a child and adolescent psychiatry clinic in Turkey. In the literature, 55% to 65% of patients presenting to child and adolescent clinics are males, and in our study this rate was found to be 62.8%. This was attributed to males reaching physiological maturity later, and psychiatric disorders are more common and more easily diagnosed since males express disease findings more [22,23].

In numerous series, the majority of patients presenting to child and adolescent clinics consisted of children in the 7–11 years age group [24,25,26]. In our study also, 45.5% of the patients were children in the 7–11 years age group. The high rate of presentations in this age group may be attributed to the emergence of learning problems, adaptation problems, and ADHD findings, because starting school and socialization take place in this period.

In this study, the most common symptoms were inattention (19.2%), hyperactivity (17.9%), irritability (13.6%), and naughtiness (11.6%), showing similarity to recent studies [8]. However, in a study conducted in Turkey in 1979, the most common causes of appointments to child and adolescent psychiatry were reported as enuresis (21.5%), stuttering (16%), intellectual disability (12%), while in another study performed in 2011, these causes were found as irritability (17.6%), naughtiness-disobedience (17.2%), and poorer school performance (16.9%) [27,28]. The main reason of this change of symptoms may be attributed to public awareness of ADHD.

Recart et al. reported the most common diagnoses in patients presenting to child and adolescent psychiatry clinics as ADHD, conduct disorder, and adaptation disorders; Harpaz-Rotem and Rosenheck as ADHD, depression, and anxiety disorders; and Ucar et al. as ADHD, depression, and anxiety disorders [8,22,25]. In our study, ADHD, SLD, and conduct disorder were the most common diagnoses. In light of this information, it is obvious that the most common psychiatric diagnosis established in child and adolescent psychiatric clinics is ADHD. ADHD is a psychiatric disorder with the highest increase of diagnosis rates in childhood, with a 3% increase recorded yearly from 1997 to 2006 [25]. Its incidence is 8.8% in the current publications, and is reported as 2.5% in adulthood [6,29]. ADHD is often associated with other psychiatric disorders [30,31]. In addition, its association has been shown with other systemic disorders, such as restless leg syndrome, sleep disturbance, and neurologic diseases [32]. These associations may be resulted from dopaminergic and norepinephrinergic imbalances that are involved in the pathophysiology and ADHD, and increased proinflammatory response [33,34]. In our study, association of ADHD and SLD, and ADHD diagnosis was established in majority of patients with epilepsy, supporting these opinions.

In our study, the frequency of SLD diagnosis was higher than in previous studies. The rates of SLD have been reported differently among studies because of the differences in diagnostic criteria. It is thought that diagnosis of SLD increased after DSM-5 SLD diagnostic criteria were changed. The incidence of SLD was reported between 5% and 15% in DSM-5 [6]. SLD and ADHD are reported as the most commonly associated psychiatric disorders [29]. There is even evidence that these two disorders are based on a common genetic ground [35]. In our study, 90% of SLD cases were accompanied by ADHD, and 23.5% of ADHD cases were accompanied by SLD, thus SLD being the second most common diagnosis was not surprising. The fact that the prevalence of ADHD and SLD with a genetic basis is high in our study suggests that there may be a genetic predisposition to these disorders in our country.

Consistent with the literature, in our study ADHD, SLD, ODD, and motor disorders were higher in males, while anxiety disorders, trauma-related disorders, depressive disorders, somatic symptoms and related disorders, impulse control disorder, adolescent problems, and eating disorders were higher in females [36].

In a study by Durukan et al. in 2011, a psychiatric diagnosis was established in 74.7% of patients who presented to child and adolescent psychiatry clinics. In that study, the rate of undiagnosed patients was 49.7% in the 0–6 age group, 17.1% in the 7–11 age group, and 13.6% in the 12–18 age group [26]. Similar to that study, in our study the rate of diagnosed patients was 79%. Our results of the rate of undiagnosed patients in the 7–11 and 12–18 age groups were similar with that study, although our rate of undiagnosed patients in the 0–6 age group was lower than that study. This difference may result from the lower number of cases (n = 533) in that study, as well as the changes in DSM-5 diagnostic criteria, and increased public awareness.

In our study, the most common symptom in the 0–6 age group was speech delay, which reflects developmental delays. In the literature, the most common cause of presentation in 7–11 and 12–18 age groups is poor school performance. In our study, the most common symptoms in these age groups were hyperactivity and inattention, respectively. This may be attributed to the increased rate of ADHD diagnosis, and increased public awareness of ADHD. Of patients in the 0–6 age group, 38.6% of patients were undiagnosed. The most commonly established diagnosis in this age group was global developmental delay, consistent with the literature [26]. Similar to our results, ADHD has been found as the most common diagnosis in the 7–11 age group. In the literature, mood disorders and substance-related and addictive disorders have been frequently reported in the 12–18 age group [26,37]. In our study, the most common diagnosis was ADHD in this age group, indicating the increased prevalence of ADHD.

In our study, the most common symptoms were inattention, irritability, and hyperactivity in patients who did not receive a psychiatric disorder. Sibling jealousy, sleep disturbance, and physical complaints were significantly higher among the undiagnosed patients. This indicates sensitivity of families for ADHD and behaviors of their children.

In the USA, the rate of presentation to mental health services raised to 13.3% between 2010 and 2012 from 9.2% between 1996 and 1998. In the same period, the use of psychotropic drugs in child and adolescent populations raised to 8.9% from 5.5% [38]. In the Chinese population, Song et al. revealed that there was a 19.2% increase in the overall use of psychotropic drugs during 2005 to 2010 [39]. A similar trend was described in Iceland, Germany, and Spain [40,41].

In our study, drug therapy was initiated in approximately two thirds (61.8%) of patients who received a psychiatric diagnosis. The use of psychotropic drugs in Turkey was reported to be 23.4% in a study conducted in 1992, 24.1% in 2005, and 43.4% in 2014 [8,9,10]. In the light of this information, our study indicated that the use of psychotropic drugs has shown an increase in Turkey as in other countries.

In our study, monotherapy was selected in 71.3% of the patients. The most commonly used drugs were psychostimulants, antipsychotics, and antidepressants, while the most common combination therapies were psychostimulant + antipsychotic and antipsychotic + atomoxetine. In three studies conducted in Turkey between 2005 and 2014, the most preferred drugs in children and adolescents were reported as antidepressants [8,9,42]. In a study conducted in 2018, similar to our study, psychostimulants were in the first rank [43]. In a study by Olfson et al. with children and adolescents in the USA, the most commonly used drugs were reported as psychostimulants (22.6%), antidepressants (13.4%), and antipsychotics (7.5%) [38]. This suggests that the preferred drug treatment changed due to the increased rate of diagnosis in ADHD. In our study, the most commonly preferred psychostimulant was methylphenidate, the most commonly selected antidepressant was fluoxetine, and the most commonly chosen antipsychotic was risperidone, suggesting that these drugs are reliable and licensed for the childhood period [44,45,46,47,48].

Remarkably, polypharmacy, which is defined as prescription of two or more drugs concurrently for a patient, was used by 28.7% in our study. The use of polypharmacy in children and adolescents is increasing in developed and developing countries [15,45,49]. Similar to other countries, the use of psychiatric polypharmacy in our country has reached significant rates. According to our study, the use of psychotropic drugs shows an increase by age. This was attributed to that an existing psychiatric disorder becomes more complex with age, and the rate of comorbidities increases. In addition, more frequent outpatient control visits in patients receiving drug therapy compared to those not receiving, and in patients using combination therapy compared to those using monotherapy suggest that drug therapy increases patient compliance.

### Limitations of the Study

The most important limitation is our study consisting of data from a single center and its retrospective design. Because this study is based on hospital record and files, the clinician may not have written or updated all the symptoms and/or diagnoses on the notes; this may have affected the results of the study. Another limitation is that since our study was conducted in a tertiary health center, our cases are likely to consist of children and adolescents with more complex clinical tables.

## 5. Conclusions

Our study shows that there is an increase in the rate of presentations to child and adolescent psychiatry outpatient clinics in Turkey, and the rate of diagnosis and initiation of drug therapy is high among presenting children. In our country, the prevalence of ADHD is increasing, and psychiatric polypharmacy has reached significant rates. Psychotropic drug prescription in Turkey should be carefully monitored in children and adolescents. Our study emphasizes that clinicians, especially children and adolescent psychiatrists, pediatricians, and family physicians, should be more careful about polypharmacy for children and adolescents and should try out non-pharmacological interventions for psychiatric disorders. Our study also emphasizes that clinicians should be more cautious about attempts to conduct non-pharmacological interventions for psychiatric disorders and for polypharmacy for children and adolescents.

## Figures and Tables

**Table 1 medicina-55-00159-t001:** Psychiatric symptoms of study population by gender.

Symptoms	Total (*n* = 2066)	Male (*n* = 1298)	Female (*n* = 768)	*p*
Inattention, *n* (%)	396 (19.2)	280 (21.6)	116 (15.1)	<0.001
Hyperactivity, *n* (%)	370 (17.9)	309 (23.8)	61 (7.9)	<0.001
Irritability, *n* (%)	282 (13.6)	173 (13.3)	109 (14.2)	0.58
Naughtiness, *n* (%)	239 (11.6)	171 (13.2)	68 (8.9)	0.003
Reading, writing, maths problems, *n* (%)	132 (6.4)	93 (7.2)	39 (5.1)	0.06
Timidness, introvert, *n* (%)	125 (6.1)	76 (5.9)	49 (6.4)	0.62
Various fears (exam, claustrophobia, loneliness etc.), *n* (%)	121 (5.9)	60 (4.6)	61 (7.9)	0.002
Speech delay, *n* (%)	120 (5.8)	95 (7.3)	25 (3.3)	<0.001
Poorer school performance, *n* (%)	109 (5.3)	63 (4.9)	46 (6.0)	0.26
Disobedience, *n* (%)	93 (4.5)	66 (3.2)	27 (3.5)	0.09
Homework and school denial, *n* (%)	83 (4.0)	53 (4.1)	30 (3.9)	0.84
Language impairment, *n* (%)	80 (3.9)	65 (5.0)	15 (2.0)	0.001
Self-mutilation, suicide attempt, *n* (%)	69 (3.3)	29 (2.2)	40 (5.2)	<0.001
Physical complaints (nausea, vomiting, fainting, etc.), *n* (%)	65 (3.1)	16 (1.2)	49 (6.4)	<0.001
Unhappiness, anhedonia, *n* (%)	58 (2.8)	20 (1.5)	38 (4.9)	<0.001
School-teacher problems, *n* (%)	56 (2.7)	46 (3.5)	10 (1.3)	0.002
Sleep disturbance, *n* (%)	55 (2.7)	22 (1.7)	33 (4.3)	0.001
Disruptive behavior (stealing, running away from home, harming friends etc.), *n* (%)	49 (2.4)	31 (2.4)	18 (2.3)	0.94
Elimination problems (enuresis, encopresis), *n* (%)	47 (2.3)	31 (2.4)	16 (2.1)	0.65
General developmental delay, *n* (%)	45 (2.2)	26 (2.0)	19 (2.5)	0.47
Headiness, *n* (%)	44 (2.1)	32 (2.5)	12 (1.6)	0.17
Obsession, *n* (%)	43 (2.1)	26 (2.0)	17 (2.2)	0.74
Forgetfulness, *n* (%)	43 (2.6)	33 (2.5)	20 (2.6)	0.93
Tic disorder, *n* (%)	42 (2.0)	35 (2.7)	7 (0.9)	0.005
Habits (finger sucking, hair pulling, nail biting), *n* (%)	38 (1.8)	15 (1.2)	23 (3.0)	0.003
Inability to make eye contact, *n* (%)	32 (1.5)	28 (2.2)	4 (0.5)	0.004
Sibling jealousy, *n* (%)	23 (1.1)	11 (0.8)	12 (1.6)	0.13
Eating disorder, *n* (%)	27 (1.3)	4 (0.3)	23 (3.0)	<0.001
Not responding to being called, *n* (%)	17 (0.8)	17 (1.3)	0 (0)	0.001
Judicial admission, *n* (%)	15 (0.7)	9 (0.7)	6 (0.8)	0.82
Family conflict, *n* (%)	15 (0.7)	6 (0.5)	9 (1.2)	0.06
Hallucination, delusion, *n* (%)	14 (0.7)	6 (0.5)	8 (1.0)	0.12
Sexual abuse, *n* (%)	7 (0.3)	0 (0)	7 (0.9)	0.001
Boredom, *n* (%)	6 (0.3)	0 (0)	6 (0.8)	0.003

**Table 2 medicina-55-00159-t002:** Psychiatric diagnosis of study population by gender.

DSM-5 Disorders	Total (*n* = 2066)	Male (*n* = 1298)	Female (*n* = 768)	*p*
ADHD, *n* (%)	808 (39.1)	616 (47.5)	192 (25)	<0.001
SLD, *n* (%)	211 (10.2)	149 (11.5)	62 (8.1)	0.01
Conduct disorder, *n* (%)	149 (7.2)	100 (7.7)	49 (6.4)	0.26
ODD, *n* (%)	142 (6.9)	112 (8.6)	30 (3.9)	<0.001
Intellectual disability, *n* (%)	142 (6.9)	84 (6.5)	58 (7.6)	0.34
Anxiety disorder, *n* (%)	134 (6.5)	63 (4.9)	71 (9.2)	<0.001
Trauma- and stressor-related disorders, *n* (%)	113 (5.5)	59 (4.5)	54 (7.0)	0.042
Communication disorders, *n* (%)	97 (4.7)	72 (5.5)	25 (3.3)	0.017
ASD, *n* (%)	94 (4.5)	75 (5.8)	19 (2.5)	0.001
Global developmental delay, *n* (%)	74 (3.6)	43 (3.3)	31 (4.0)	0.39
Depressive disorders, *n* (%)	64 (3.1)	17 (1.3)	47 (6.1)	<0.001
Motor disorders, *n* (%)	57 (2.8)	45 (3.5)	12 (1.6)	0.011
Elimination disorders, *n* (%)	52 (2.5)	37 (2.9)	15 (2.0)	0.2
Obsessive-compulsive and related disorders, *n* (%)	41 (2.0)	20 (1.5)	21 (2.7)	0.06
Somatic symptom and related disorders, *n* (%)	20 (1.0)	4 (0.3)	16 (2.1)	<0.001
Sleep-wake disorders, *n* (%)	20 (1.0)	9 (0.7)	11 (1.4)	0.09
Impulse control disorder, *n* (%)	18 (0.9)	6 (0.5)	12 (1.6)	0.009
Adolescent problems, *n* (%)	17 (0.8)	5 (0.4)	12 (1.6)	0.004
Bipolar disorders, *n* (%)	8 (0.4)	4 (0.3)	4 (0.5)	0.48
Substance related and addictive disorders, *n* (%)	7 (0.3)	5 (0.4)	2 (0.3)	0.63
Feeding and eating disorders, *n* (%)	7 (0.3)	0 (0)	7 (0.9)	0.001
Schizophrenia, *n* (%)	5 (0.2)	2 (0.2)	2 (0.3)	0.37
Grief reaction, *n* (%)	4 (0.2)	1 (0.1)	3 (0.4)	0.12
Internet addiction, *n* (%)	3 (0.1)	2 (0.2)	1 (0.1)	1
Gender dysphoria, *n* (%)	3 (0.1)	2 (0.2)	1 (0.1)	1
Other, *n* (%)	8 (0.4)	4 (0.4)	4 (0.5)	0.48
No child psychiatric diagnosis recorded, *n* (%)	433 (21.0)	240 (18.5)	193 (25.1)	<0.001

Abbreviations: ADHD; attention-deficit hyperactivity disorder, ASD; autism spectrum disorder, ODD; oppositional defiant disorder, SLD; specific learning disorder, DSM-5; fifth edition of the Diagnostic and Statistical Manual of Mental Disorders.

**Table 3 medicina-55-00159-t003:** The most common symptoms and psychiatric diagnoses according to age groups.

0–6 years (*n* = 466)	7–11 years (*n* = 941)	12–18 years (*n* = 659)
**Symptoms, *n* (%)**		
Speech delay, 92 (19.7)	Hyperactivity, 244 (25.9)	Inattention, 141 (21.4)
Naughtiness, 83 (17.8)	Inattention, 243 (25.8)	Irritability, 133 (20.2)
Hyperactivity, 53 (11.4)	Irritability, 121 (12.9)	Hyperactivity, 73 (11.1)
Language impairment, 32 (6.9)	Naughtiness, 107 (11.4)	Various fears, 65 (9.9)
Timidity, 31 (6.7)	Learning disability, 103 (10.9)	Naughtiness, 49 (7.4)
Inability to make eye contact, 25 (5.4)	Timidity, 66 (7.0)	Poorer school performance, 46 (7.0)
**Psychiatric Diagnosis, *n* (%)**		
No child psychiatric diagnosis, 180 (38.6)	ADHD, 528 (56.1)	ADHD, 261 (39.6)
Global developmental delay, 68 (14.6)	No child psychiatric diagnosis, 155 (16.5)	No child psychiatric diagnosis r, 98 (14.9)
Communication disorders, 59 (12.7)	SLD, 149 (15.8)	Conduct disorder, 73 (11.1)
Trauma- and stressor-related disorders, 49 (10.5)	ODD, 95 (10.1)	Intellectual disability, 64 (9.7)
ASD, 32 (6.9)	Intellectual disability, 76 (8.1)	SLD, 61 (9.3)
ODD, 23 (4.9)	Conduct disorder, 63 (6.7)	Anxiety disorder, 61 (9.3)

Abbreviations: ADHD; attention-deficit hyperactivity disorder, ASD; autism spectrum disorder, ODD; oppositional defiant disorder, SLD; specific learning disorder.

**Table 4 medicina-55-00159-t004:** Clinical characteristics of children with and without psychiatric diagnosis.

Symptoms	Diagnosed	No Diagnosis	*p*
Inattention, *n* (%)	349 (21.4)	47 (10.9)	<0.001
Hyperactivity, *n* (%)	338 (20.7)	32 (7.4)	<0.001
Irritability, *n* (%)	229 (14.0)	53 (12.2)	0.33
Sibling jealousy, *n* (%)	12 (0.7)	11 (2.5)	0.001
Timidness, introvert, *n* (%)	96 (5.9)	29 (6.7)	0.52
Sleep disturbance, *n* (%)	34 (2.1)	21 (4.8)	0.001
Various fears (exam, claustrophobia, loneliness etc.), *n* (%)	96 (5.9)	25 (5.8)	0.93
Physical complaints (nausea, vomiting, fainting, etc.), *n* (%)	42 (2.6)	21 (5.3)	0.004
Poorer school performance, *n* (%)	99 (6.1)	10 (2.3)	0.002
Reading, writing, maths problems, *n* (%)	124 (7.6)	8 (1.8)	<0.001
Speech delay, *n* (%)	107 (6.6)	13 (3.0)	0.005
Disobedience, *n* (%)	85 (5.2)	8 (1.8)	0.003
Homework and school denial, *n* (%)	72 (4.4)	11 (2.5)	0.078
Language impairment, *n* (%)	75 (4.6)	5 (1.2)	0.001
Self-mutilation, suicide attempt, *n* (%)	64 (3.9)	5 (1.2)	0.004
Unhappiness, anhedonia, *n* (%)	46 (2.8)	12 (2.8)	0.95
School-teacher problems, *n* (%)	50 (3.1)	6 (1.4)	0.056
Disruptive behavior (stealing, running away from home, harming friends etc.), *n* (%)	46 (2.8)	3 (0.7)	0.01

**Table 5 medicina-55-00159-t005:** Clinical characteristics of study population according to drug treatment status.

Some Variables	With Pharmacotherapy (*n* = 1009)	Without Pharmacotherapy (*n* = 624)	*p*
Age, years	11.6± 3.6	8.8± 4.6	<0.001
Gender, male, *n* (%)	680 (67.4)	378 (60.6)	0.22
Psychiatric outpatient visit	5.4± 3.2	2.7± 2.0	<0.001
ADHD, *n* (%)	695 (68.9)	113 (18.1)	<0.001
Conduct disorder, *n* (%)	127 (12.6)	22 (3.5)	<0.001
Anxiety disorder, *n* (%)	68 (6.5)	66 (10.9)	0.002
Specific learning disorder, *n* (%)	160 (15.9)	51 (8.2)	<0.001
Oppositional defiant disorder, *n* (%)	91 (9.0)	51 (8.2)	0.55
Mental retardation, *n* (%)	92 (9.1)	50 (8.0)	0.44
Trauma- and stressor-related disorders, *n* (%)	32 (3.2)	81 (13.0)	<0.001
Communication disorders, *n* (%)	13 (1.3)	84 (13.5)	<0.001
Autism spectrum disorder, *n* (%)	73 (7.2)	21 (3.4)	0.001
	**Monotherapy (*n* = 724)**	**Combination therapy (*n* = 291)**	
Age, years	11.6± 3.7	11.4± 3.4	0.48
Gender, male, *n* (%)	459 (63.4)	224 (77.0)	<0.001
Admission number	5.0±2.8	6.5± 3.7	<0.001
ADHD, *n* (%)	442 (61.0)	253 (86.9)	<0.001
Conduct disorder, *n* (%)	52 (7.2)	75 (25.8)	<0.001
Anxiety disorder, *n* (%)	58 (8.0)	8 (2.7)	0.002
SLD, *n* (%)	113 (15.6)	47 (16.2)	0.83
ODD, *n* (%)	48 (6.6)	43 (14.8)	<0.001
Intellectual disability, *n* (%)	48(6.6)	44 (15.1)	<0.001
Trauma- and stressor-related disorders, *n* (%)	30 (4.1)	2 (0.7)	0.002
ASD, *n* (%)	45 (4.4)	28 (2.8)	0.057

Abbreviations: ADHD; attention-deficit hyperactivity disorder, ASD; autism spectrum disorder, ODD; oppositional defiant disorder, SLD; specific learning disorder.

**Table 6 medicina-55-00159-t006:** Distribution of psychotropic medications in the study population (*n* = 1015).

Drugs	*n*	%
**Monotherapy**		
Long-acting psychostimulants	298	29.3
Short-acting psychostimulants	49	5.0
Atomoxetine	92	9.0
Antidepressants	162	16.0
Antipsychotics	158	24.0
Antihistaminics	27	2.6
Mood stabilizers	7	0.7
**Combination therapy**		
Psychostimulant + Antipsychotic	122	12.0
Antipsychotic +Atomoxetine	45	4.4
Long acting + short acting psychostimulant	23	2.3
Psychostimulant + Atomoxetine	15	1.5
Antipsychotic + Mood stabilizer	7	0.7
Antipsychotic + Antidepressant	5	0.5
Psychostimulant + Antipsychotic + Atomoxetine	23	2.3
Psychostimulant + Antipsychotic + Mood stabilizer	10	1.0
Psychostimulant + Antipsychotic + Atomoxetine	9	0.9
Antipsychotic + Antidepressant + Mood stabilizer	6	0.6
Antipsychotic + Atomoxetine + Mood stabilizer	9	0.9

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
