# Peer review of "Current Pattern of Psychiatric Comorbidity and Psychotropic Drug Prescription in Child and Adolescent Patients"

_medicina, 2019, doi:10.3390/medicina55050159_

Reviewer 1 Report

This is an interesting topic on the rate of presentation and diagnosis of mental disorders in child and adolescent and psychiatry outpatient clinical diagnosis and medication prescription in Turkey, a country with a very special and unique cultural and geographical situation. But I think that there are some major and minor shortcomings with the study which need to be addressed.

·         Major:

1.    I think a paragraph on Turkey is worth to be added at the beginning of the study indicating its special demographics.

2.    The introduction part needs to be updated. I think a more extensive and updated review of literature is needed. This part might be more focused around the need for doing similar studies and the impacts these types of reports might have on the service providers.

3.    I think it is important to talk about the approach which is done in doing the diagnosis of the mental problems. Was it solely based on the DSM5 criteria or any diagnostic tools were engaged in the process? I think using diagnostic tools for determining conditions such as ADHD and ASD is inevitable and it is not advised to do the diagnosis solely based on DSM’s criteria.

4.    Since both increased mental diagnosis (SLD and ADHD) are based in the genetic ground, how do you justify this finding?

5.    It is strongly recommended to add a paragraph on the applicability of the conclusion. What is the benefit of the presented finding? What group of the readers are aimed?

6.    Why the presented finding compared with the USA data? What is the indication for this comparison? Is it possible to do a comparison with countries with a more similar economical and geographical or political condition?

·          Minor:

•    Line 46 (Psychiatric disease profile in children and adolescents shows quite difference from adulthood) it is not clear what the authors mean by using this sentence and what type of special features they are pointing to. Therefore the sentence is not clear and needs to be rewritten.

•    I have also difficulty to understand the justification which is given in line 47 to 50  (Recently, significant 48 changes have been made in psychiatric diagnosis criteria of these disorders). The cited reference clearly talks about a change which has been made to DSM5 but this missed in the text and the authors talk about general diagnostic changes which need more references.

•    Line 80 has a punctuation problem (analyzed by chi-square or Fisher’s exact test, where applicable.. Statistical significance was accepted 80 as p<0.05.)< span="">

•    In the results part I think it is useful to report standard deviation (SD) for the age group as well as the mean score (Of the 2066 patients, 1298 (62.8%) were male and the mean age was 10.14±4.42 years of all patients, 22.6% were in 0-6 years, 45.5% in 7-11 years, and 31.9% in 12-18 years age groups.)

•    Reword sentence number 199 (In our study, different for other studies diagnosis of SLD was prominent.)

Author Response

Reviewer 1:

Comments and Suggestions for Authors

This is an interesting topic on the rate of presentation and diagnosis of mental disorders in child and adolescent and psychiatry outpatient clinical diagnosis and medication prescription in Turkey, a country with a very special and unique cultural and geographical situation. But I think that there are some major and minor shortcomings with the study which need to be addressed.

  Major:

1.    I think a paragraph on Turkey is worth to be added at the beginning of the study indicating its special demographics.

We added to this paragraph introduction sectironTurkey has a growing young population, Turkey's health ministry last 10 years primarily serves awareness activities for psychiatric disorders such as ASD and SLD. There were also significant increases in the number of child and adolescent psychiatrists. As a result of these, the number of admissions to child and adolescent psychiatry has been increased in recent years. The use of psychotropic drugs in children and adolescents in Turkey was reported as 23.4% in a study conducted in 1992, 24.1% in 2005, and 43.4% in 2014”

2.    The introduction part needs to be updated. I think a more extensive and updated review of literature is needed. This part might be more focused around the need for doing similar studies and the impacts these types of reports might have on the service providers.

We uptated  introduction section.

3.    I think it is important to talk about the approach which is done in doing the diagnosis of the mental problems. Was it solely based on the DSM5 criteria or any diagnostic tools were engaged in the process? I think using diagnostic tools for determining conditions such as ADHD and ASD is inevitable and it is not advised to do the diagnosis solely based on DSM’s criteria.

We added information about evaluation tools “Files of all patients were examined in details, and patients’ demographic characteristics, symptoms of admission, psychiatric diagnoses psychotropic drugs initiated, and side effect profiles were recorded. Psychiatric diagnoses of the patients were based on DSM-5 criteria using clinician's opinion and supportive assessment tools. For example these assessment tools included Modified Checklist for Autism in Toddlers for ASD, Turgay ADHD Evaluation Scale for ADHD, Specific Learning Difficulties Symptom Scale for SLD, State-Trait Anxiety Inventory  for anxiety disorder, Beck depression scale for depressive disorder.”

4.    Since both increased mental diagnosis (SLD and ADHD) are based in the genetic ground, how do you justify this finding?

Although there are reports that these two disorders may be of genetic origin, it cannot be claimed that all SLD and ADHD cases are genetic. In our study, we can not make a causal inference about the increase of these two disorders in our country. however, we can argue that genetic susceptibility can be attributed to our country. Indeed, in our country, consanguineous marriages are an important problem and prepare the ground for genetic diseases.

We added this sentence “The fact that the prevalence of ADHD and SLD with genetic basis is high in our study suggests that there may be a genetic predisposition to these disorders in our country.”

5.    It is strongly recommended to add a paragraph on the applicability of the conclusion. What is the benefit of the presented finding? What group of the readers are aimed?

We added this paragraph conclusion section “. Psychotropic drug prescription in Turkey should be carefully monitored in children and adolescents. Our study emphasizes that clinicians, especially children and adolescent psychiatrists, pediatricians and family physicians , should  be more careful about polypharmacy for children and adolescents to try out non-pharmacological  interventions for psychiatric disorders. Our study also  emphasizes that clinicians should be more cautious about attempts to conduct non-pharmacological interventions for psychiatric disorders and for polypharmacy for children and adolescents.

.”

 6.    Why the presented finding compared with the USA data? What is the indication for this comparison? Is it possible to do a comparison with countries with a more similar economical and geographical or political condition?

In the literature, comprehensive data  on this  area are mostly based on USA data. However, as the reviewer stated, the demographic characteristics of our country differ from USA.  We added some other countries' data to the discussion section. You can follow below

In the USA, rate of presentation to mental health services raised to 13.3% between 2010-2012 from 9.2% between 1996-1998. In the same period, the use of psychotropic drug in child and adolescent population raised to 8.9% from 5.5% [37]. In Chinese population, Song et al revealed that there was a 19.2% increase in the overall use of psychotropic drugs during 2005 to 2010 [38].  A similar trend was described in Iceland, Germany and Spain [39,40].Our study, drug therapy was initiated in about two third (61.8) of patients who received a psychiatric diagnosis. The use of psychotropic drugs in Turkey was reported as 23.4% in a study conducted in 1992, 24.1% in 2005, and 43.4% in 2014 [8-10]. In the light of this information, our study indicated that the use of psychotropic drugs has shown increase in Turkey as in the other contries” 

Minor:

 •    Line 46 (Psychiatric disease profile in children and adolescents shows quite difference from adulthood) it is not clear what the authors mean by using this sentence and what type of special features they are pointing to. Therefore the sentence is not clear and needs to be rewritten.

We changed the sentence to “The common psychiatric disorders such as mood ,  stress related and somatoform disorders  affecting adults also affect many child and  adolescents [3,4]. However, ADHD, SLD, autism spectrum disorder (ASD), conduct disorder, and elimination disorders such as enuresis and encopresis are seen especially in child and adolescent period [5, 6]. Recently, diagnostic criteria of many psychiatric disorders were modified in DSM-5 [6].”

 •    I have also difficulty to understand the justification which is given in line 47 to 50  (Recently, significant 48 changes have been made in psychiatric diagnosis criteria of these disorders). The cited reference clearly talks about a change which has been made to DSM5 but this missed in the text and the authors talk about general diagnostic changes which need more references.

We changed the sentence “Recently, diagnostic criteria of many psychiatric disorders were modified in DSM-5 [6].”

•    Line 80 has a punctuation problem (analyzed by chi-square or Fisher’s exact test, where applicable.. Statistical significance was accepted 80 as p<0.05.)< strong="">

We removed one of the points.

•    In the results part I think it is useful to report standard deviation (SD) for the age group as well as the mean score (Of the 2066 patients, 1298 (62.8%) were male and the mean age was 10.14±4.42 years of all patients, 22.6% were in 0-6 years, 45.5% in 7-11 years, and 31.9% in 12-18 years age groups.)

In the study, the percent of age groups was specified. it is not appropriate to specify the SD of the proportion of age groups (this is the opinion of the statistical expert)

 •    Reword sentence number 199 (In our study, different for other studies diagnosis of SLD was prominent.)

We changed the sentence to” In our study, the frequency of SLD diagnosis was higher than in previous studies”

Reviewer 2 Report

Abstract, line 20 - Having read the manuscript, I would argue that differences by age (group) are not explored (and/ or not reported), systematically as for gender so would place less (no) emphasis on it in the abstract.

Abstract, line 23 - We could guess the most commonly presenting age group is 7-11 from the mean (SD).  Stating this is wasting words on more valuable findings.

Abstract - the results do not say anything about gender differences despite you emphasising it in the methods.  

Line 47 - define SLD in the text.

Line 65 - a study cannot be retrospective and cross-sectional.  Having read the manuscript, I think you mean it is a retrospective cohort.

Line 78 - which groups?

Line 85 - was the number of psychiatrist visits Normally distributed?  If so give standard deviation.  If not, then give median and (lower quartile, upper quartile).  Also round the number of visits to the nearest whole number as it is not possible to have 0.9 of a visit.

Line 86 and elsewhere - do you mean admission or do you mean psychiatric outpatient visit?

Lines 89-90 - It would be easier to comprehend and more in line with the aims of the study if you told us the percentage who were taking at least one psychotropic drug (it is taken as given that the other were not taking any).

Table 2 - is mental retardation an accepted term?  If not, please change to something more acceptable.

Line 130 - 41.6 is not majority.  Also this is not needed as it can be seen that this is likely to be the biggest group from the mean age of those without a diagnosis.

Line 147 - boy ratio.  Do you mean boy: girl ratio?

Lines 153-154 - repetition from earlier in the results, remove.

Line 266 - If the data were gained through notes, then there would be no recall bias.  The only bias would be from the clinician not writing all the medications and/ or diagnoses in the notes. 

Author Response

Reviewer 2:

Abstract, line 20 - Having read the manuscript, I would argue that differences by age (group) are not explored (and/ or not reported), systematically as for gender so would place less (no) emphasis on it in the abstract.

 We re-edited abstract section

 Abstract, line 23 - We could guess the most commonly presenting age group is 7-11 from the mean (SD).  Stating this is wasting words on more valuable findings.

We removed this sentence in abstract section “The most common age of presentation was 7-11 years age group.”

 Abstract - the results do not say anything about gender differences despite you emphasising it in the methods. 

 We added to these sentences abstract section. “The most common symptoms of admission were hyperactivity (23.8%) and inattention (21.6%) in males, inattention (15.1%),  and irritability (14.2%) in females, and 79% of the patients received one or more psychiatric diagnosis. The most common psychiatric diagnoses in both gender were attention-deficit hyperactivity disorder (ADHD)  specific learning disorder (SLD) and conduct disorder , respectively.”

Line 47 - define SLD in the text.

 SLD was defined in abstract section

 Line 65 - a study cannot be retrospective and cross-sectional.  Having read the manuscript, I think you mean it is a retrospective cohort.

 We changed study disagn to “retrospective cohort study”

 Line 78 - which groups?

 We change this sentence to “Mean differences between continuous measures  were compared by Student's t-test” and Continuous measures include age and psychiatric outpatient visit.

Line 85 - was the number of psychiatrist visits Normally distributed?  If so give standard deviation.  If not, then give median and (lower quartile, upper quartile).  Also round the number of visits to the nearest whole number as it is not possible to have 0.9 of a visit.

We stated this information “The mean number of psychiatric interviews of the patients within one year was found as 4.0±3.0.”

Line 86 and elsewhere - do you mean admission or do you mean psychiatric outpatient visit?

 We change to “psychiatric outpatient visit” all text

 Lines 89-90 - It would be easier to comprehend and more in line with the aims of the study if you told us the percentage who were taking at least one psychotropic drug (it is taken as given that the other were not taking any).

 We changed this sentence to “One or more psychotropic therapy was initiated in 49.1% of the patients.”

Table 2 - is mental retardation an accepted term?  If not, please change to something more acceptable.

 We changed in text and all tables to “Intellectual disability

 Line 130 - 41.6 is not majority.  Also this is not needed as it can be seen that this is likely to be the biggest group from the mean age of those without a diagnosis.

 We removed this sentence  “and majority of children (41.6%) were in 0-6 years age group. “

 Line 147 - boy ratio.  Do you mean boy: girl ratio?

 This only refers to the male gender dominace. Thus, this word changed to “male gender predominance”

 Lines 153-154 - repetition from earlier in the results, remove.

We removed this sentence

 Line 266 - If the data were gained through notes, then there would be no recall bias.  The only bias would be from the clinician not writing all the medications and/ or diagnoses in the notes.

 This information has been added to study limitation

Round  2

Reviewer 1 Report

This is an improved version of the previous paper. I think the authors have done their best to respond to most of the comments. But I think there is still one issue which needs to be addressed. In respond to the comment on the diagnosis it is said that:

For example these assessment tools included Modified Checklist for Autism in Toddlers for ASD, Turgay ADHD Evaluation Scale for ADHD, Specific Learning Difficulties Symptom Scale for SLD, State-Trait Anxiety Inventory  for anxiety disorder, Beck depression scale for depressive disorder.”

As it is clearly cited in the references M-CHAT is a screening tool, not a diagnostic one and it is applicable with toddlers and could not cover the age range which is mentioned in the present paper! We have some other gold standard scales such as ADI-R, ADOS and some other diagnostic scales ( I am not so sure if any Turkish version is available for these scales but as far as I am informed there is a Turkish version of GARS2 (Gilliam Autism Rating Scale translated and standardized by Ibrahim H. Diken and Autism Behavior Checklist ABC which scale which has been translated by the same researcher) which should be used for the diagnosis of ASD in its different forms and level of severity. This needs to be clarified. The other scales which are mentioned also need to be checked if they are used as a screening or diagnostic tool. 

Author Response

This is an improved version of the previous paper. I think the authors have done their best to respond to most of the comments. But I think there is still one issue which needs to be addressed. In respond to the comment on the diagnosis it is said that:

For example these assessment tools included Modified Checklist for Autism in Toddlers for ASD, Turgay ADHD Evaluation Scale for ADHD, Specific Learning Difficulties Symptom Scale for SLD, State-Trait Anxiety Inventory  for anxiety disorder, Beck depression scale for depressive disorder.”

As it is clearly cited in the references M-CHAT is a screening tool, not a diagnostic one and it is applicable with toddlers and could not cover the age range which is mentioned in the present paper! We have some other gold standard scales such as ADI-R, ADOS and some other diagnostic scales ( I am not so sure if any Turkish version is available for these scales but as far as I am informed there is a Turkish version of GARS2 (Gilliam Autism Rating Scale translated and standardized by Ibrahim H. Diken and Autism Behavior Checklist ABC which scale which has been translated by the same researcher) which should be used for the diagnosis of ASD in its different forms and level of severity. This needs to be clarified. The other scales which are mentioned also need to be checked if they are used as a screening or diagnostic tool.

This is absolutely true, the gold standard for diagnosis in the autism spectrum disorder is still considered to be the clinical diagnosis. In addition to the clinical interview, screening tests M-CHAT (for under 3years)   and ABC (for over 3 years) were used. We added to Autism Behavior Checklist, and checked other assesmnet tools.

Reviewer 2 Report

Please proofread the paper for English language.

Line 35 - do you mean prevalence not incidence?

line 113 and elsewhere - remove the word admission.  You are not describing hospital admissions, you are describing outpatient visits/ appointments/ consultations.  In most cases the sentence reads ok without the word admission.  For example, on line 147, the sentence could start with the word Symptoms.

Line 123 - change statistically to significantly

Line 149-150 and elsewhere - 38.6% is not the majority.  The majority is over half.  You mean the most common symptom/ diagnosis/ prescribed drug etc.

Line 285 and following - I am not sure selected is the right word.

Line 288-289 - Is the reasoning for these drugs being used more is that they are licenced for use in children?

Line 302 - The point is that the notes may be incomplete due to clinicians not updating the notes during/ after a given appointment.  It is impossible to know if this is the case.  It is likely symptoms and/ or diagnoses rather than medications are missing as medications are arguably more important and electronic notes may be linked to prescribing.

Author Response

Reviewer  2

 Please proofread the paper for English language.

The English language of the manuscript has been revised and edited by native speaker.

 Line 35 - do you mean prevalence not incidence?

Actually, the prevalence is more suitable because it includes all cases referred to the outpatient clinic, not just new diagnosis

We changed the incidence to prevalance

 line 113 and elsewhere - remove the word admission.  You are not describing hospital admissions, you are describing outpatient visits/ appointments/ consultations.  In most cases the sentence reads ok without the word admission.  For example, on line 147, the sentence could start with the word Symptoms.

In the direction of the reviewer’s criticism, we have removed or changed this word all text.

 Line 123 - change statistically to significantly

We changed the statistically to significantly in this line

 Line 149-150 and elsewhere - 38.6% is not the majority.  The majority is over half.  You mean the most common symptom/ diagnosis/ prescribed drug etc.

We changed this term

 Line 285 and following - I am not sure selected is the right word.

We changed to this sentence to This suggests that the preferred drug treatment changed due to increased rate of diagnosis in ADHD”

 Line 288-289 - Is the reasoning for these drugs being used more is that they are licenced for use in children?

These drugs licensed for childhood period, we added this information and related references.

 Line 302 - The point is that the notes may be incomplete due to clinicians not updating the notes during/ after a given appointment.  It is impossible to know if this is the case.  It is likely symptoms and/ or diagnoses rather than medications are missing as medications are arguably more important and electronic notes may be linked to prescribing.

This is absolutely true, since we added to this sentence  conclussion section “Because this study is based on hospital record and files, the clinician may not have written or updated all the symptoms and / or diagnoses on the notes.  it may have affected the results of the study.
